# Perspectives of wheelchair users with spinal cord injury on fall circumstances and fall prevention: A mixed methods approach using photovoice

Hardeep Singh[1,2], Carol Y. Scovil[1,3], Geoff Bostick[4], Anita Kaiser[1,2,5], B. Catharine Craven[1,2,6,7], Susan B. Jaglal[1,2,7,8], Kristin E. Musselman[1,2,8¤]*

1 KITE, Toronto Rehab-University Health Network, Toronto, ON, Canada, 2 Rehabilitation Sciences Institute, Faculty of Medicine, University of Toronto, Toronto, ON, Canada, 3 Dept. of Occupational Science & Occupational Therapy, Faculty of Medicine, University of Toronto, Toronto, ON, Canada, 4 Dept. of Physical Therapy, University of Alberta, Edmonton, Canada, 5 Canadian Spinal Research Organization, Toronto, ON, Canada, 6 Division of Physical Medicine & Rehabilitation, Faculty of Medicine, University of Toronto, Toronto, ON, Canada, 7 Institute of Health Policy, Management & Evaluation, University of Toronto, Toronto, ON, Canada, 8 Dept. of Physical Therapy, Faculty of Medicine, University of Toronto, Toronto, ON, Canada

¤ Current address: SCI Mobility Lab, KITE, Toronto Rehab-University Health Network, Toronto, ON, Canada
* Kristin.Musselman@uhn.ca

**Data Availability Statement:** Data cannot be shared publicly because data contain potentially identifying information. Data are available from the

## Abstract

### Introduction

Wheelchair users with spinal cord injury are at a high risk of falls. However, the perspectives of wheelchair users with spinal cord injury on their fall circumstances and their preferences for fall prevention strategies/interventions remain understudied. Therefore, we aimed to: a) describe the circumstances of falls experienced by wheelchair users with spinal cord injury over a six-month period, b) explore their perspectives of why falls occurred in certain situations, and c) explore their perspectives on recommended content/structure of fall prevention strategies/interventions.

### Methods

This sequential explanatory mixed methods study had two phases. Phase I involved tracking of falls experienced by wheelchair users with spinal cord injury over six months, in which participants completed a survey after experiencing a fall to track the number/circumstance of each fall. Data from the surveys were descriptively reported. Phase II involved a photovoice focus group discussion of the survey findings and their preferences for fall prevention strategies/interventions. Data from the focus group discussion were analyzed using a thematic analysis.

### Results

Thirty-two participants completed phase I. More than half of the participants fell at least once in six months. Falls commonly occurred in the afternoon during a transfer, or when participants were wheeling over uneven ground. One-third of the falls caused an injury. Eleven

University Health Network Ethics Committee (contact via email: reb@uhnresearch.ca) for researchers who meet the criteria for access to confidential data.

**Funding:** This study was funded by a Craig H. Neilsen Foundation Psychosocial Research Grant to KEM, a Canadian Institutes of Health Research Fellowship to HS and Toronto Rehabilitation Institute Student Scholarship to HS. SBJ holds the Toronto Rehabilitation Institute Chair-University Health Network at the University of Toronto. The funders did not have a role in the study's design, data collection and analysis, decision to publish or preparation of the manuscript.

**Competing interests:** The authors have declared that no competing interests exist.

participants that fell during phase I participated in the focus group. Two main themes were identified from the discussion: 1) "circumstances surrounding the falls" (e.g. when falls occurred, the home is a 'safe space') and 2) "suggestions and preferences for fall prevention strategies/interventions" (e.g. fall prevention involves all, fall prevention training available as needed).

## Conclusion

Fall prevention strategies/interventions should be an integral component of rehabilitation practices across the lifespan. Participants recommend customizing fall prevention strategies/interventions to their specific needs to guide the structure, content, and delivery of targeted fall prevention programs.

## Introduction

Falls are a complex, multifactorial "emerging public health crisis" [1–3]. A fall from a wheelchair can result in injuries and, in some cases, mortality for a wheelchair user [1, 3–8]. Further, the aftermath of a fall can have lasting psychosocial impacts on the individual who fell, including greater dependence on others, activity restrictions, and negative emotions (e.g. fear and anxiety) [3, 9–11]. A fall not only impacts the person who experiences it, but can also have a lasting and traumatic impact on caregivers/family members/friends whom witness the fall/aftermath of a fall [11].

Based on a recent systematic review and metanalysis, 69% (95% CI 60–76%) of wheelchair users with spinal cord injury (SCI) experience a fall each year [1]. Falls and fall risk among wheelchair users with SCI are often caused by multiple interacting factors including: biological (e.g. leg/core spasm), behavioural (e.g. distraction), social (e.g. error by a caregiver/family member/friend), economic (e.g. costly adaptive equipment), and environmental factors (e.g. uneven ground) [1, 3, 12, 13]. A history of falls is a strong predictor of future falls among wheelchair users with SCI [4]. To complicate matters, the causes of fall risk and falls are dynamic, meaning that a wheelchair user's risk of falling can change over time [13]. Furthermore, a wheelchair user's fall risk is dependent on their individual circumstances (e.g. their home and community environment, daily activities, and comorbidities) [13].

The high incidence of falls and the potential for a serious fall-related injury among wheelchair users with SCI [1, 4, 7] indicate the need for more effective fall prevention strategies/interventions. However, the complexity of a wheelchair user's falls and fall risk, create significant challenges for how to effectively reduce or ideally, prevent, falls from occurring. In two previous studies, people with SCI reported that the fall prevention training they received was insufficient for their dynamic fall prevention needs, as their physical abilities and function changed over time [9, 13]. Yet, there is a paucity of evidence-based fall prevention strategies/interventions to prevent/reduce falls and injurious falls among this population group [6, 7, 14]. Recently, Rice & colleagues [15] investigated the effectiveness of a "brief fall prevention intervention" that aimed to decrease the number and concern of falls among wheelchair users with SCI. The program consisted of one session of educational and practical training to improve transfer skills and postural control, followed by a 12-week period of participants' implementing the intervention in their home/community. Although this intervention was feasible and demonstrated potential to reduce fall incidence, the authors noted a need to increase

participants' engagement in the intervention [15]. Existing fall prevention strategies/interventions are a good starting point for fall prevention, but more research is needed to effectively meet the complex fall prevention needs and preferences of wheelchair users with SCI [13]. A thorough understanding of each person's situations, concerns and perceptions of what solutions may be viable, may enhance the utility of fall prevention interventions targeting this population.

While prior studies have looked at falls using quantitative surveys [4, 5, 7], the depth of information that was collected using surveys alone, is limited [1, 16]. The circumstances surrounding falls experienced by wheelchair users with SCI remain unclear [12] and there is a need to collect first person experiences in the form of qualitative data (e.g. semi-structured interviews with people with SCI regarding reasons for falls and fall prevention) along with quantitative data (e.g. number of falls) to fully understand the quantitative findings. For example, a survey can assess the breadth of the circumstances surrounding falls; however, in order to generate customized fall prevention programs for an individual/specific condition, there is a need to explain fall circumstances in more depth to understand the idiosyncrasies of each situation. Therefore, we selected a mixed methods approach to advance existing knowledge of fall circumstances and explore fall prevention from the perspective of wheelchair users living with SCI. Addressing these knowledge gaps can guide the development of future fall prevention strategies/interventions that are more suitable for the targeted population. The specific aims of our study were to: 1) describe the circumstances for falls experienced by wheelchair users with SCI over a six-month tracking period; 2) explore their perspectives on why falls occurred in certain situations; and, 3) explore their perspectives on the content and structure of fall prevention strategies/interventions.

## Materials and methods

### Design

This was a six-month longitudinal, sequential explanatory mixed methods study with two phases (Fig 1) [17]. A mixed methods approach was suitable to investigate the complex issue of

**Phase 1 (quantitative):**

**Phase 2 (qualitative):**

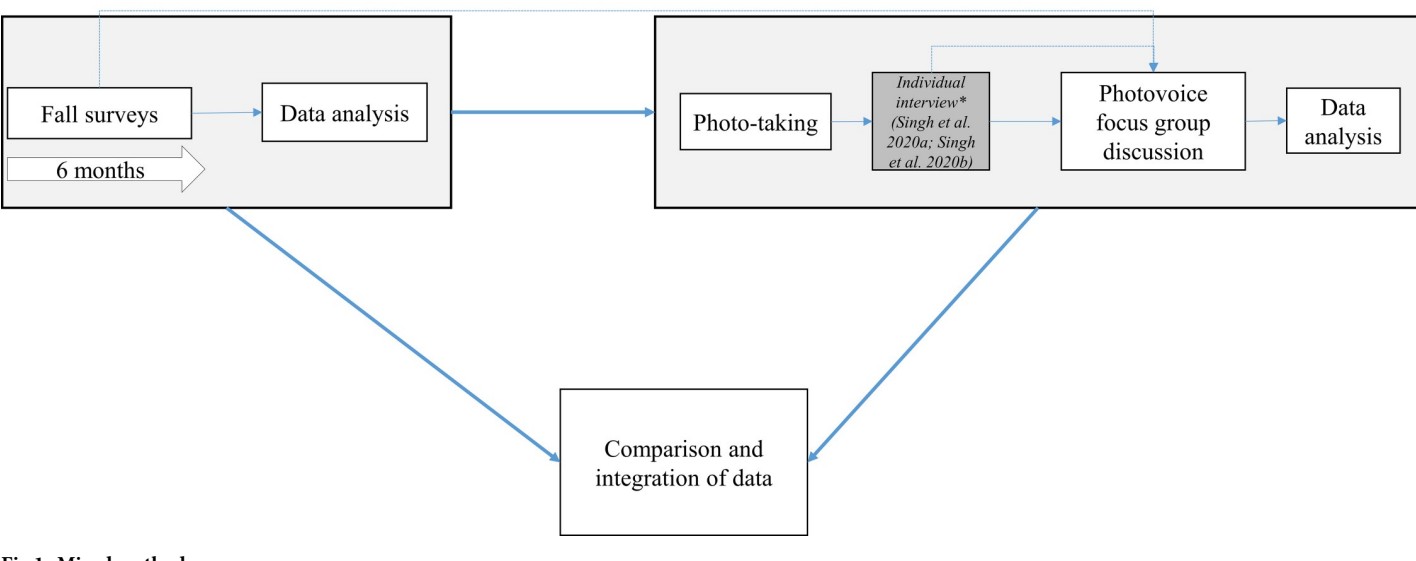

**Fig 1. Mixed methods process.**

falls, as this approach allows the researcher to combine the strengths of both quantitative and qualitative methods to yield comprehensive results [18]. For instance, the quantitative methods in phase I were used to examine trends in the circumstances surrounding falls, while the qualitative methods in phase II were used to interpret the trends and identify the participants' perspectives on suggestions and preferences for fall prevention strategies/interventions. Photovoice is a collaborative research technique that can generate valuable insights to inform the development of patient-centered interventions [19]. Furthermore, our pilot work demonstrated that photovoice was a feasible approach to gaining an in-depth understanding about falls in individuals with SCI [9].

## Settings and participants

This study stems from a larger project that examined falls in 65 participants with SCI (32 were wheelchair users and 33 were ambulatory individuals) [20]. This research occurred at a tertiary SCI rehabilitation centre in Toronto, Canada—the Lyndhurst Centre, Toronto Rehabilitation Institute-University Health Network (UHN). The Research Ethics Board of the UHN and the Health Sciences Research Ethics Board of the University of Toronto provided ethical approval for this study. All participants provided written and verbal consent to participate in this study. Individuals were eligible to participate in phase I of the current study if they met the following pre-defined eligibility criteria: (i) had a chronic (≥one year) traumatic SCI, (ii) had a SCI between C1-L1 [American Spinal Injury Association Impairment Scale (AIS) grade A-D], (iii) lived in the community for ≥one month, (iv) used a manual or power wheelchair for ≥four hours per day [21], and, (v) were ≥18 years of age. Information about this study was shared with the public via the following methods: recruitment flyers posted through SCI Ontario [22] and in rehabilitation clinics, and using the Lyndhurst Centre's central recruitment database [23]. Also, consenting participants were asked to share the study details with their peers.

Inclusion in phase II required participants to have: (i) completed phase I and (ii) experienced ≥ one fall during the six-month tracking period. We approached 15 individuals from phase I who were eligible for phase II; 11 of the 15 individuals agreed to participate in phase II. This was an appropriate size for the study methods given samples of seven to ten individuals are recommended for a photovoice focus group [24, 25].

## Data collection

**Phase I.**   Demographics and impairment characteristics were collected from each participant at the start of the six-month tracking period. Participants were provided with the following written and verbal definition of a fall: "an event which results in a person coming to rest inadvertently on the ground or floor or other lower level" [26]. Participants were instructed to complete an electronic (Qualtrics Survey Software, Dallas, TX, USA) or paper-based survey, each time they fell during the six-month tracking period [27] (see S1 File for the fall survey). The surveys were used to document the number and circumstances of each fall (e.g. time, location, activity and perceived cause). In order to reduce recall errors, participants were instructed to complete the survey within 24 hours of experiencing a fall. A research team member called each participant every three to four weeks during the tracking period to remind participants to complete the fall survey if/when they fell, maintain the participants' engagement in the study, and assist with survey completion if assistance was required.

**Phase II.**   Phase II involved a two-hour long photovoice focus group discussion. Photovoice is a qualitative methodology that enables participants to critically reflect on their lived experiences [28]. The photovoice process involved participants (1) completing a photo-assignment on an issue or topic, (2) voicing their individual experiences in a one-on-one interview,

(3) discussing their collective experiences in a group discussion, and then (4) sharing the information with key stakeholders in order to influence programs/policies [24, 25]. Prior to this focus group, all 11 participants completed a photo-assignment and one-on-one interview with the primary author. The photo-assignment asked participants to capture photographs of situations or things that influenced their fall risk and how the risk of falls impacted their work and recreational activities. The interviews explored the participants' individual experiences of fall risk and the psychosocial impact of falls. These findings were previously reported in two publications [11, 13].

In phase II of the current study, we conducted the photovoice focus group discussion [24, 25]. The focus group was facilitated by a licensed occupational therapist and PhD candidate (HS), a Rehabilitation Engineer and SCI Knowledge Mobilization Specialist (CYS), and a licensed physical therapist, Assistant Professor, and Scientist (KEM). During the focus group, HS presented trends from the surveys (from phase I) and invited participants to share their interpretations of the data (e.g. responses from the surveys showed that most falls occurred at home, what are your thoughts on this?). Next, each participant selected one photograph from their photo-assignment that highlighted a situation or thing that influenced their fall risk or the impact of a fall or risk of fall. The intent of photo sharing was to promote dialogue [24] about falls/fall prevention and for group members to collectively reflect on their fall prevention needs. This was followed by discussions about key themes uncovered in the previous conducted photo-elicitation interviews [11, 13], and aspects of fall prevention (e.g. from your perspective what does an ideal fall prevention training/intervention look like?) (see S2 File for the focus group interview guide). As recommended in photovoice methodology, the focus group discussion ended with a discussion about how the findings could be disseminated [24].

## Data analysis

**Phase I.**   Data from the surveys were entered in Microsoft Excel (Microsoft Corporation) for management and analysis. Descriptive data from the surveys were presented by response frequency to each question, medians with ranges, and means with one standard deviation (SD). Responses were also examined for males and females separately. Responses to the open-ended survey questions were descriptively reported. Independent t-tests were used to compare age and time since injury between fallers and non-fallers.

**Phase II.**   The focus group discussion was audio-recorded and transcribed verbatim by HS. The transcript was uploaded onto NVivo 12 (QSR International Pty Ltd., Burlington, MA), which is a software used to organize qualitative data. An inductive thematic analysis was used to analyze the focus group data. An inductive approach allowed us to interpret themes that were data-driven and closely reflected the participants' viewpoints [29]. HS read the transcript multiple times to become immersed in the data. HS and KEM independently conducted initial coding of the transcript. HS and KEM compared their interpretations of the initial codes. HS created the preliminary themes based on patterns/similarities and differences among the codes. After this, KEM reviewed and verified that the themes reflected the transcript content. The final themes were formed based on joint input from transcript reviewers.

## Rigour and credibility

The following aspects were incorporated into the study design to enhance trustworthiness of the findings [30–34]. First, credibility of the findings was enhanced with involvement of two authors who independently coded the data. In addition, quotes from participants were used to support our interpretations from the qualitative data. Second, our use of two data collection methods (i.e. surveys and a focus group discussion) allowed us to confirm that our

interpretations closely reflected the perspectives of the participants (i.e. method triangulation). Third, we outlined each researcher's role in the data collection/analysis. In addition, we provided information about the researchers' professional background and qualifications because the researchers are the major instruments of data collection/analysis in qualitative research [33]. Fourth, we followed the "Good Reporting of A Mixed Methods Study" guidelines to increase the quality of this mixed methods research (see S3 File for the guidelines) [35]. Lastly, we provided a detailed description of our methods and participants' characteristics.

## Results

### Phase I

Thirty-two participants (17 male and 15 female) participated in phase I (see Table 1 for the demographics and impairment characteristics of the participants). Of the participants, five (four fallers, one non-faller) ambulated short distances within their home. Fallers tended to be younger than non-fallers (mean age 46.2±12.2 years versus 53.7±7.6 years, respectively, p = 0.049).

During the six-month tracking period, 29 falls were reported. Of the entire sample, 53.1% (n = 17) of wheelchair users had at least one fall (8 male, 9 female). Of those that fell, 50.0% (n = 8) fell more than one time. Among manual wheelchair users, 56.5% (n = 13, 4 male, 9 female) experienced at least one fall. Of the manual wheelchair users who fell, 38.5% (n = 5)

**Table 1. Demographic characteristics and impairment characteristics of participants followed in phase I.**

| | All Participants (n = 32) | All fallers (n = 17) | Non-fallers (n = 15) | Fallers using manual wheelchair (n = 13) | Fallers using power wheelchair (n = 4) | Non-fallers using manual wheelchair (n = 10) | Non-fallers using power wheelchair (n = 5) |
|---|---|---|---|---|---|---|---|
| **Mean age ± 1 SD** (years) | 49.7±10.8 | 46.2±12.2 | 53.7±7.6 | 43.2±12.2 | 55.8±6.5 | 53.5±6.4 | 54.2±10.5 |
| **Median age** | 51.5 | 49 | 54 | 49 | 57 | 54 | 56 |
| **Min-max** (years) | 21–69 | 21–62 | 41–69 | 21–57 | 47–62 | 44–62 | 41–69 |
| **Male** (n) | 17 | 8 | 9 | 4 | 4 | 6 | 2 |
| **Female** (n) | 15 | 9 | 6 | 9 | 0 | 4 | 3 |
| **Years living with SCI** | | | | | | | |
| Median | 27 | 29 | 27 | 22 | 32 | 24.5 | 27 |
| Min-max | 3–63 | 4–45 | 3–63 | 6–40 | 4–45 | 3–63 | 22–51 |
| **Mechanism of injury** | | | | | | | |
| Motor vehicle accident (n) | 17 | 9 | 8 | 9 | 0 | 5 | 3 |
| Sports accident (n) | 11 | 6 | 5 | 3 | 3 | 3 | |
| Fall (n) | 4 | 2 | 2 | 1 | 1 | 2 | |
| **Level of SCI** | | | | | | | |
| Cervical | 17 | 8 | 9 | 4 | 4 | 4 | 5 |
| Thoracic | 13 | 7 | 6 | 7 | 0 | 6 | 0 |
| Lumbar | 2 | 2 | 0 | 2 | 0 | 0 | 0 |
| **Motor complete** (AIS A or B): | 26 | 12 | 14 | 8 | 4 | 9 | 5 |
| **Motor incomplete** (AIS C or D): | 6 | 5 | 1 | 5 | 0 | 1 | 0 |

SD: standard deviation. AIS: American Spinal Cord Injury Association Impairment Scale.

fell more than once. Forty-four percent (n = 4, all males) of power wheelchair users experienced at least one fall, with 75.0% (n = 3) of the fallers having more than one fall.

The majority of falls occurred in the afternoon or evening either inside or just outside of the home. Falls commonly occurred during a transfer or while a participant performed an activity from their wheelchair (e.g. going over uneven ground, sports/exercise). Sports/exercise included activities such as "exercising from my chair", "catching a frisbee" and "playing wheelchair basketball". When considering activities by the location, more than half of falls in the home and nearly two third of falls in the community occurred during a transfer, changing positions or while a participant was going over uneven ground, an incline, or through a doorway. See Table 2 for the circumstances of the falls experienced by participants.

Fig 2 contains a summary of the causes of all falls reported by all participants. When considering falls by location (i.e. in the home versus community; see Table 2), the most frequently reported causes of falls inside and just outside the home were: moving too quickly, "other" factors (e.g. leaned too far, error by an assistant, new equipment, and not wearing footwear), and having poor balance. The most frequently reported causes of falls in the community included: being distracted, tripping, and "other" factors (e.g. assistant error, leaned too far back).

When comparing fall circumstances among females and males, females experienced more falls at night and during a transfer or while they were wheeling over uneven ground than males; males did not report any falls while wheeling over uneven ground. In addition, females experienced falls while they were in the outside in the community, whereas males did report any falls in this location.

## Fall-related injuries

Ten fall-related injuries were reported by nine participants who experienced one or more falls that resulted in an injury, with the most common injuries being cuts/scrapes and pain. Of these nine participants, five were recurrent fallers and four fell only one time over the six-month period.

More than half of the injurious falls occurred inside or just outside the home environment; one of which resulted in a broken bone ("right shoulder") when a manual wheelchair user was changing positions (i.e. leaning forward while seated in his wheelchair to retrieve an item from the floor inside the home). Four of the reported injurious falls occurred in the community, with three of these falls requiring medical assistance/assessment.

## Phase II

Demographic characteristics of the 11 participants that participated in phase II are displayed in Table 3.

Two themes were identified from the focus group discussion: 1) Circumstances surrounding the falls and 2) Suggestions and preferences for fall prevention (see Table 4 for the themes, subthemes, and supporting quotes).

## Theme 1: Circumstances surrounding the falls

**Subtheme 1: When falls occurred.** Wheelchair users believed that the majority of falls occurred in the afternoon and evening as these "are the more active periods in the day" (P4). P4 explained not getting enough sleep the night before could increase a person's fall risk during the afternoon/evening time. Participants also believed that visual disturbances caused by poor lighting in the evening could be a potential factor contributing to these falls: "[it is] hard to see when the sun goes down and it gets really dark outside" (P5).

**Table 2. Fall circumstance for entire group, power wheelchair users and manual wheelchair users.**

| Fall circumstance | All fallers | Manual wheelchair users | Power wheelchair users |
|---|---|---|---|
| Total number of falls | 29 | 22 | 7 |
| **Time of fall (%, n*)** | | | |
| Afternoon | 11 (37.9%) | 8 (36.4%) | 3 (42.9%) |
| Evening | 10 (34.5%) | 9 (40.9%) | 1 (14.3%) |
| Night | 5 (17.2%) | 5 (22.7%) | 0 |
| Morning | 3 (10.3%) | 0 | 3 (42.9%) |
| **Location of fall (%, n*)** | | | |
| Home indoors | 14 (48.3%) | 10 (45. 5%) | 4 (57.1%) |
| Community outdoors | 7 (24.1%) | 7 (31.8%) | 0 |
| Home outdoors | 4 (13.8%) | 2 (9.1%) | 2 (28.6%) |
| Community indoors | 4 (13.8%) | 3 (13.6%) | 1 (14.3%) |
| Work indoors | 0 | 0 | 0 |
| Work outdoors | 0 | 0 | 0 |
| **Activity during falls (%, n*)** | | | |
| Getting in/out of bed, vehicle, or bath/shower | 7 (24.1%) | 5 (227%) | 2 (27.6%) |
| Going over uneven ground | 4 (13.8%) | 4 (18.2%) | 0 |
| Sports/exercise | 4 (13.8%) | 4 (18.2%) | 0 |
| ☆ Changing positions | 3 (10.3%) | 2 (9.1%) | 1 (14.3%) |
| Going through doorway | 2 (6.9%) | 0 | 2 (27.6%) |
| Walking | 2 (6.9%) | 2 (9.1%) | 0 |
| Going up or down an incline | 2 (6.9%) | 1 (4.5%) | 1 (14.3%) |
| Assisted standing | 1 (3.4%) | 1 (4.5%) | 0 |
| Dressing | 1 (3.4%) | 1 (4.5%) | 0 |
| Opening drawer | 1 (3.4%) | 1 (4.5%) | 0 |
| Taking photo | 1 (3.4%) | 1 (4.5%) | 0 |
| Walking up or down the stairs | 1 (3.4%) | 0 | 1 (14.3%) |
| **Causes of falls by location** _Inside or just outside the home (%, n**)_ | | | |
| Moving too quickly | 7 (14.0%) | 4 (12.5%) | 3 (16.7%) |
| Other | 7 (14.0%) | 5 (15.6%) | 2 (11.1%) |
| Poor balance | 6 (12.0%) | 4 (12.5%) | 2 (11.1%) |
| Weakness in legs | 5 (10.0%) | 4 (12.5%) | 1 (5.6%) |
| Legs gave out | 5 (10.0%) | 4 (12.5%) | 1 (5.6%) |
| Distracted | 3 (6.0%) | 1 (6.3%) | 2 (11.1%) |
| Multitasking | 3 (6.0%) | 1 (3.1%) | 2 (11.1%) |
| Slipped | 2 (4.0%) | 1 (3.1%) | 1 (5.6%) |
| Spasm | 2 (4.0%) | 2 (3.1%) | 0 |
| Tired | 2 (4.0%) | 1 (3.1%) | 1 (5.6%) |
| Not using mobility aid/safety equipment | 2 (4.0%) | 0 | 2 (11.1%) |
| Dark/poorly lit environment or problems with vision | 2 (4.0%) | 2 (6.3%) | 0 |
| Tripped | 1 (2.0%) | 1 (3.1%) | 0 |
| Weather | 1 (2.0%) | 1 (3.1%) | 0 |
| 'Don't know' | 1 (2.0%) | 1 (3.1%) | 0 |
| Illness | 1 (2.0%) | 0 | 1 (5.6%) |
| **_Community (%, n^)_** | | | |
| Distracted | 3 (17.6%) | 3 (18.8%) | 0 |
| Tripped | 3 (17.6%) | 3 (18.8%) | 0 |

_(Continued)_

**Table 2.** (Continued)

| Fall circumstance | All fallers | Manual wheelchair users | Power wheelchair users |
|---|---|---|---|
| Other | 3 (17.6%) | 2 (12.5%) | 1 (100%) |
| Moving too quickly | 2 (11.8%) | 2 (12.5%) | 0 |
| Poor balance | 1 (5.9%) | 1 (6.25%) | 0 |
| Tired | 1 (5.9%) | 1 (6.25%) | 0 |
| Multitasking | 1 (5.9%) | 1 (6.25%) | 0 |
| Dark/poorly lit environment | 1 (5.9%) | 1 (6.25%) | 0 |
| Alcohol | 1 (5.9%) | 1 (6.25%) | 0 |
| Weather | 1 (5.9%) | 1 (6.25%) | 0 |
| Type of injury (%, n^^) | | | |
| Cuts/Scrapes | 6 (31.6%) | 3 (23.1%) | 3 (50.0%) |
| Pain | 6 (31.6%) | 4 (30.8%) | 2 (33.3%) |
| Bruises | 3 (15.8%) | 3 (23.1%) | 0 |
| Bumped Head | 2 (10.5%) | 1 (7.7%) | 1 (16.7%) |
| Broken Bone | 1 (5.3%) | 1 (7.7%) | 0 |
| Spasm in legs | 1 (5.3%) | 1 (7.7%) | 0 |

Legend: ☆ changing positions: "Getting into and out of a body position and moving from one location to another, such as rolling from one side to the other, sitting, standing, getting up out of a chair to lie down on a bed, and getting into and out of positions of kneeling or squatting" [36]

*denominator is total number of falls reported

**denominator is total number of causes for falls inside or just outside the home

^ denominator is total number of causes for falls in the indoor/outdoor community

^^ denominator is total number of injuries reported

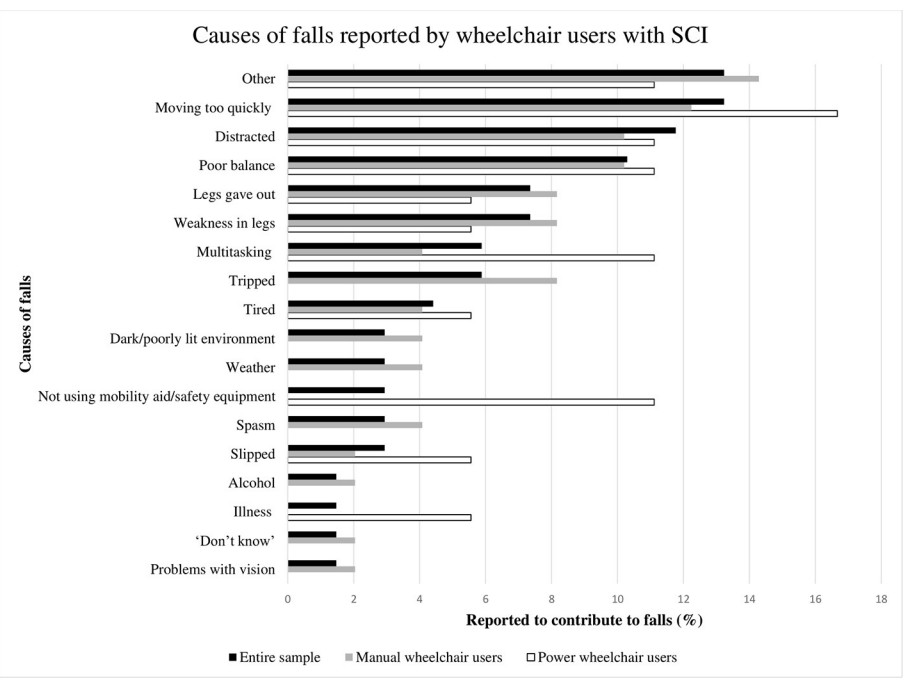

**Fig 2. Causes of falls reported by the entire sample, manual wheelchair users and power wheelchair users.**

Table 3. Demographic and impairment characteristics of participants from phase II.

| Participant code | Age (years) | Sex | Type of wheelchair | Time since injury (years) | Neurological level of injury | American Impairment Scale |
|---|---|---|---|---|---|---|
| P1 | 56 | Male | Power | 40 | C4 | C |
| P3 | 47 | Male | Power | 3 | C5 | B |
| P4 | 41 | Female | Manual | 16 | T12 | A |
| P5 | 37 | Female | Manual | 19 | T1 | A |
| P6 | 25 | Male | Manual | 6 | C7 | C |
| P7 | 48 | Female | Manual | 36 | T12 | B |
| P8 | 48 | Female | Manual | 29 | L1 | C |
| P9 | 52 | Female | Manual | 34 | T9 | B |
| P10 | 33 | Female | Manual | 15 | T12 | C |
| P11 | 61 | Male | Power | 44 | C4 | B |
| P12 | 33 | Female | Manual | 5 | T10 | D |
| | *Median (min-max)*: 47 (25–61) | | | *Median (min-max)*: 19 (3–44) | | |

**Subtheme 2: The home is a 'safe space'.** Some participants were initially surprised to see that the majority of falls recorded during phase I occurred in the home. Though, after reflection, the finding resonated with the participants. Participants asserted that carelessness, multitasking, and not paying attention to the surroundings were common causes of falls in the home. People were less vigilant about avoiding falls in their home. P4 reasoned, "usually when we are in the home, we tend to be careless. . .I pay less attention to my surroundings when I am at home because it is a 'safe space'."

P1's explanation of why falls tended to occur in the home was simply because the home was where most of the time was spent, especially if an individual was not employed. Consistent with the survey data from phase I, participants reasoned that falls in the home were related to performing a greater number of transfers into and out of their wheelchair and spending more time outside of the wheelchair while at home. P8 detailed her recent experience of "nearly falling" while transferring out of her shower. P8 asserted that equipment failure was the underlying cause in that situation:

> On the shower bench I have one of those [anti-skid bath bench cover]. . .I went to transfer out of my shower today into my chair and it just flew. . .equipment can just fail on you. . .you're used to doing things a million times, and you do it, but stuff fails on you, you can just fall.

Bathing was perceived to be a risky activity for many participants, which primarily occurred in the home. P6 described losing his balance when he experienced paresthesia and having a muscle spasm while sitting in his shower chair: "I sat on my shower chair before and my legs had gone to sleep. Normally I feel [my legs] but they had gone to sleep which also makes my muscles kind of turn off and I threw a hip flexion spasm which pulled my upper body forward. . . it's not preventable because you don't know with a muscle spasm." P1 added that the location of a spasm was important to consider. Spasms in the obliques "pull you sideways", but the spasms that were "right in the hip flexors. . .pull you forward."

Based on the results from phase I, some participants fell while performing activities while they were upright (e.g. walking, standing in a standing frame, and climbing stairs). These participants ambulated short distances in their home (primarily for exercise) and/or used a standing frame for therapy/exercise. After reflecting on falls that occurred while he was ambulating,

**Table 4. Themes, subthemes, and supporting quotes.**

| Theme 1: Circumstances surrounding the falls | |
|---|---|
| **Subthemes** | **Supporting quote** |
| **Subtheme 1:** When falls occurred | "This morning in the shower, it was one of the worst wipeouts that I've ever had." (P8)<br>"If you haven't slept well then the afternoon could be quite difficult." (P4) |
| **Subtheme 2:** The home is a 'safe space' | "When you're at home, you're more likely to do things like multitasking…like if you're rushing to get out the door and get to work or get things done. And that goes back to not paying attention because at home you might be rushing more, and you may have accidents." (P8) |
| **Subtheme 3:** Situations causing falls in the community | "There is a lot of uneven terrain outside. So you might think you're fine because you're on a flat surface and then all of the sudden there is a crack in the sidewalk or a dip or pothole and you don't notice because you're not paying attention and then fall. I have fallen out of my chair that way a few times." (P6) |
| **Theme 2: Suggestions and preferences for fall prevention strategies/interventions** | |
| **Subthemes** | **Supporting quote** |
| **Subtheme 1:** Fall prevention: an integral piece of new and existing rehabilitation practices | "Nobody asked me 'do you want the bigger casters?' or even that we have an option for them." (P8)<br>"I think providing those options because you can customize a wheelchair to the end of time…They should have kind of a guide…And like kinda provide the pros and cons about each [safety feature] and give you the option because unless you actually say I want bigger casters, often times [the wheelchair prescriber] won't even ask you." (P12)<br>"You know trying to do a wheelie until you can't do it anymore and end up on the floor." (P6) |
| **Subtheme 2:** Fall prevention involves all | "I think OTs should be there, as well as hearing from people who have been using wheelchairs." (P10)<br>"My friend… a Rugby player…put mats around me and taught me a whole bunch of different controlled falls." (P12)<br>"I think we need [fall prevention education for] the PSWs or therapists." (P6)<br>"[People need to understand] that falls will happen, and they will not be the end of you. It's not something that needs to be told just to the person [with SCI] but also to the family and the support person." (P4) |
| **Subtheme 3:** Receiving knowledge from multiple avenues | "I would do it over the Internet and do it in a crowdsourcing type of manner as well…Then people can share their ideas and experiences as well." (P6)<br>"An online program is great, but you have to have hands-on." (P4)<br>"Opening up [fall prevention information] to the consumers, so people in wheelchairs could use the information." (P6) |
| **Subtheme 4:** Fall prevention strategies/ intervention available as needed | "Not necessarily invite everyone to one session but if you had it ongoing, it would be more successful." (P11)<br>"It should be multiple times. Get outside and try to wheel as much as you can. You have to remove that fear and anxiety." (P12) |

P1 explained: "when we ambulate, sometimes the muscle will fire, or not fire, and all of the sudden, we are on the ground and it's without warning."

**Subtheme 3: Situations causing falls in the community.** Since fewer falls were reported by participants in the community, there was consensus that public spaces were generally becoming more accessible since the introduction of the provincial accessibility mandates. However, participants expressed conflicting views about the new "accessible" crosswalk design.

From P5's perspective, the new crosswalk design could potentially prevent falls: "on top of the sidewalk. . .[there is] a metal sheet with raised circular designs so that the friction and resistance is really well managed there. . .I think that's something that's heading in the right direction and will potentially prevent falls in the future." However, P8 argued that the design inhibited her mobility and was a fall hazard for her: "actually, I find that inhibiting. . .a lot of the curbs are steep going up and when you have the metal plate and you're trying to push up and over it, and with snow and ice on it, it's actually more resistance and more of an obstacle and barrier. . .going down them and up them can throw you off balance."

Although it was not captured in the surveys, many participants had experienced falls or near falls related to equipment failure/malfunction in the community. P8 recalled her wheelchair's caster fell off while she was in a museum: "I was wheeling through a museum and my front caster flew out across the floor and my whole chair tipped forward. Stuff like that happens." In another example, a few manual wheelchair users explained that the device they used to assist with wheelchair propulsion sometimes malfunctioned. P7 recalled nearly falling when her device malfunctioned: "[The technology] can just run by itself. I have hit the wall and I hit the glass door. . .I was having coffee and sitting at the table and it just started by itself."

Another contributor to falls in the community that was not well reflected in the surveys, was assistance from members of the public. This was sometimes problematic as members of the public "have no clue about balance points" (P8).

## Theme 2: Suggestions and preferences for fall prevention strategies/interventions

Participants discussed ideas that could guide the development of fall prevention strategies and interventions. According to P12, "everyone is sort of different" in terms of their function and mobility. As such, a single approach to fall prevention would not sufficiently address the varying fall prevention needs/preferences of every wheelchair user. In order to effectively manage various fall risk factors encountered by a wheelchair user with SCI, there was a general consensus that an individualized fall prevention approach would be most appropriate.

**Subtheme 1: Fall prevention: An integral piece of new and existing rehabilitation practices.** As falls were caused by multiple factors, participants believed that fall prevention should be a priority in new (e.g. extending fall prevention to family/caregivers, fall prevention interventions) and existing rehabilitation practices (e.g. wheelchair prescription, transfer training).

*Prioritizing fall prevention during wheelchair prescription*. Minimizing one's fall risk should be an integral aspect of wheelchair prescriptions. When prescribing a wheelchair, a therapist must consider the fit of a wheelchair as well as discuss the safety features that could reduce their risk of falling. Based on their past experiences, participants felt information about safety features and options for the wheelchair were not consistently discussed by the wheelchair prescribers. For example, prior to this focus group, P7 was not aware that large size casters could reduce fall risk and that they were an option for her wheelchair.

*Extending fall prevention education to caregivers/family members/friends*. Participants were interested in receiving information about common causes of falls and fall prevention strategies/interventions. They also believed this would be valuable for their caregivers, family members, and/or friends. Accessible vehicle drivers would also benefit from education/training on fall prevention during a vehicle transfer. Based on a past experience of falling while he was being assisted out of a cab, P1 explained it was also important for a wheelchair user to learn "how to tell a cab driver or someone how to disassemble my wheelchair or how to position my legs."

P5 explained, "the first time you fall people freak out." Fall prevention education provided to caregivers/family members/friends should consist of "healthy conversations about how falls are going to be a part of your life and how they can be managed" (P5). More specifically, participants believed that caregivers/family members/friends should be educated on what they could do if a wheelchair user fell and how they could assist the person back into their wheelchair. For example, a strategy used by P3 was to "maintain eye contact, use a calm voice. . .If you panic, they panic. . . if you're not clear with your instructions, they really don't know what to do and you want to avoid injury. You want to tell them please don't move this or that. I've fractured my leg. Can you call 911."

*Developing skills to fall correctly*. In terms of the content of fall prevention, participants emphasized the importance of learning how to fall in a way that minimizes injury within a controlled and safe setting. P10 explained, "I've had two actual falls from a chair. . .And I've done it wrong every time. . .It would be nice to learn the proper way." P8 believed learning how to recover from a fall was valuable. For her, the experience of falling in a safe setting reduced her fear of falls:

> I think having the falls in a safe environment with the physio and showing people what it's like to fall out of your chair can be helpful. It can help you learn what to do and how to hold the chair and what to do with your body to help yourself get back up. . . I found having that as my first real fall with my physio next to me helped me. I was like okay I'm not going to die. . .[the physio] was there to show me what to do and make sure I am safe.

According to participants, wheelchair users should know that the correct way to fall in order to minimize the risk of an injury was "not to put [your] wrists out" (P5). Breaking a fall with an outstretched hand was the wrong way to fall as it could lead to a wrist fracture and that could severely restrict a person's function. When falling forward, participants discussed the right way to fall was to twist the body to fall in a way that they hit their shoulder first while protecting their head, hands, and legs. Falling backwards involved participants leaning forward so that their back would hit the ground rather than their head. Another participant explained a strategy to avoid an injury to her lower extremity was to protect her knees "When I am falling forward, I move my side of my body out so that I'm not hurting my knees because that's where most of my fractures happen" (P12).

*Using technology for seeking assistance after a fall*. Although some participants recalled having a discussion with a healthcare professional about how to get up from a fall, the strategies recommended by the therapists were not always applicable or feasible in real circumstances. P12 recalled learning a fall recovery strategy; the impracticality of which sparked laughter in the group: "I was never taught how to get up into my chair if I fell. The only thing I was told by a private physiotherapist was to start putting books, if you are near a bookshelf, under each of your butt cheeks until you get up to the level that you are able to get up."

The more practical/feasible strategies that participants used to maximize their safety were mostly self-developed. For example, after a past traumatic fall, P8 carried her phone with her everywhere she went, including in the shower in case she needed to call for assistance after a fall: "Two in the morning I fell in the winter on an icy ramp. I was out there alone. It was freezing cold. I was out there so long that I thought I was going to die there." Participants highlighted the importance of having multiple means to seek assistance after a fall. For example, another participant described a situation where she was unable to access her phone. Alternatively, she used the Google home device to call for assistance.

*Advocating for safety*. During the focus group discussion, participants discussed falls that occurred due to community hazards and poorly trained accessible vehicle drivers; these

contributing factors were not reflected in the surveys. They discussed the importance of knowing who a wheelchair user could contact to request removal of a community fall hazard as well as advocating for themselves with accessible vehicle drivers. For instance, some accessible vehicle drivers were not well trained on how to provide assistance to a wheelchair user while they performed a transfer. It was evident that not all participants were aware they could contact a city official on these matters. As a result of this discussion, participants believed that all wheelchair users should be aware of how to advocate for the removal of community fall hazards.

*Transfer skills*. All participants agreed that learning individualized transfer skills was an important part of fall prevention because "a huge amount of falls happen during transfers" (P6). Participants highlighted the importance of learning transfer skills from someone who had experience using a wheelchair and could individualize the skill to them. Based on a past experience, P12 explained the therapist may not know how to customize the skills to a wheelchair user: "the therapist was trying to teach me one specific way to do [the transfer] and I didn't feel comfortable because in certain areas my proprioception would go way off. . .if you have someone who standing, that's never been in a chair trying to teach you the skills, you can sort of look at them and say that is not going to work you know? The logistics of it. . . [the therapist] was really stuck on that this is how you do it."

*Advancing wheelchair skills*. According to the participants, fall prevention should be integrated into teaching advanced wheelchair skills, such as navigating community environments (e.g. poorly maintained curbs, performing a wheelchair wheelie, etc.). Simply practicing skills in an accessible gym was "useless" when a wheelchair user was being taught advanced wheelchair skills. To ensure safety when acquiring advanced wheelchair skills, participants found practicing wheelchair skills with a safety strap was useful. In addition, P12 described receiving assistance from a peer to acquire advanced wheelchair skills: "[my friend] just spent one day when we were in training camp to teach me the different skills. . .my friend put mats around me and taught me a whole bunch of different controlled falls, going out from the side of my chair, the front and going back."

Although participants were experienced wheelchair users, it still took them some time to become familiar/comfortable with a new wheelchair. Some believed practical training could facilitate this learning process and reduce their fall risk. Becoming familiar with a new wheelchair involved learning the new wheelchair's tipping points and centre of gravity.

**Subtheme 2: Fall prevention involves all.**   Participants believed fall prevention should include wheelchair users with SCI, as well as their caregivers/family members/friends. Occupational therapists and physical therapists were seen as valuable resources to learn wheelchair skills from in the early stages of their injury. Participants explained that the therapist could only teach the basic "textbook version" of wheelchair and transfer skills. Other experienced wheelchair users with SCI were considered valuable resources to learn advanced wheelchair skills. Unlike therapists, peers with SCI had firsthand experience using a wheelchair and could offer more practical advice on fall prevention and wheelchair skills than most therapists. P6 explained, "Some OTs, they will jump into the chair and they will get to know it and they will do wheelies and all that, but there are a lot of OTs that have never sat in a chair themselves or even tried to navigate little obstacles, and so they are very disconnected from the clientele that they are working with."

**Subtheme 3: Receiving knowledge from multiple avenues.**   Participants believed that fall prevention strategies/interventions should be available through multiple formats, including verbal, written, and Internet-based. Internet-based resources were considered "more accessible" and "you could have access to it anytime, anywhere" (P3). Internet-based resources could serve those that no longer had access to rehabilitation through public/private insurance and people who did not reside close to a rehabilitation facility.

However, to acquire fall prevention skills, participants believed Internet-based resources alone were not enough. Practical "hands-on" training was needed to acquire skills to reduce/prevent falls. While some participants would be more comfortable in one-on-one training, others saw the value of a group learning environment. The advantages of learning from peers in a group setting were even evident in the current group. For example, P7 alluded to learning new skills from others during this focus group, "I have been using a wheelchair for almost 20 years, but I never received any training or teaching on how to fall. Like [P7 referred to another participant in the focus group] talked about so many skills and I was like [de-identified] she is so knowledgeable. It was my first time to know something about that."

**Subtheme 4: Fall prevention strategies/interventions available as needed.** Perceptions of when the ideal time was to receive fall prevention strategies/interventions varied among participants. Some participants believed the best time to receive fall prevention strategies/interventions was while they were in rehabilitation. P8 explained it was important to practice wheelchair skills as much as possible in order to prevent falls: "It's important in removing that fear and anxiety and you need to practice, practice, practice. So how long should it be? Every weekend you are in rehab here. All those 8 to 12 weeks that you are in here." Other participants disagreed as more urgent priorities should be addressed in rehabilitation and readiness was a barrier to learning about fall prevention.

> I think initially after my surgeries, I would have been so reluctant to do that. . .I would have always been afraid of reinjuring myself again but maybe a few months down the road when I am feeling a little more comfortable in my chair you know when I would be over the initial shock of the life changing. . .so that you're able to absorb what someone is trying to teach (P12).

Similarly, in P6's opinion, "a follow-up in a years' time" would be appropriate "because people might have encountered something out in the community that they are having an issue with where they didn't have the problem when they were in rehab because everything is fully accessible." Many argued that if the training was optional and available as needed, that could potentially address their changing fall prevention needs.

## Discussion

In this study we described the circumstances of falls experienced by wheelchair users with SCI over six months. In addition, we explored the perspectives of wheelchair users with SCI on fall circumstances and fall prevention. As there is limited research on fall prevention in this population, rehabilitation professionals lack the resources and information on how to approach fall prevention for individuals with SCI [37]. The results of this study may guide the development of future fall prevention initiatives.

We found that 53% of wheelchair users fell at least one time over a six-month period. Most fall-related injuries reported on the surveys were minor. However, previous research suggests that even minor or non-injurious falls can have significant psychosocial impacts on the lives of some wheelchair users with SCI [11].

Despite being experienced wheelchair users, participants commonly experienced a fall during a transfer and while wheeling. As such, participants in our study believed fall prevention should be an integral aspect of wheelchair prescription, and wheelchair and transfer skill training. Previous studies have found that many community-dwelling wheelchair users with SCI do not have advanced-level wheelchair skills [38–40]. A wheelie (i.e. balancing on rear wheels) is an important skill as it allows a wheelchair user to negotiate curbs and confined spaces, as well

as safely and independently navigate their environment [41]. Yet, many wheelchair users cannot perform a wheelie [42]. Reasons for this are that a wheelchair user may be fearful of losing their balance and/or the clinician teaching the skill may lack confidence/knowledge of the skill [42].

In Canada, wheelchair skills training is predominantly delivered by occupational therapists [43]. An investigation into the current practices of wheelchair skills training in Canadian rehabilitation centres revealed that advanced-level wheelchair skills training was insufficient for optimal wheeled mobility. The insufficiency results from multiple barriers, including the therapists not being adequately trained in advanced wheelchair skills, a lack of time/resources in rehabilitation, and an overall shorter length of stay in rehabilitation [43]. Advanced wheelchair skills may potentially reduce the number of tips and falls out of the wheelchair and fall-related injuries [43]. In order for therapists to deliver effective training of advanced wheelchair skills, therapists should receive education and hands-on training of advanced wheelchair skills in their entry-to-practice education [43].

Although participants considered their home environments as a "safe space", most falls occurred inside or just outside the home. Similar to previous studies [3, 12, 13], we found falls were multifactorial. The challenges with existing fall prevention approaches applied in practice is that they fail to consider the multifactorial nature of falls [13]. For example, home assessments are commonly recommended to reduce fall risk for community-dwelling wheelchair users [44], but these assessments tend to focus only on environmental hazards [45]. Although environmental factors in the home (e.g. home modifications, assistive devices) are important considerations for fall prevention [12], the interaction of the other components such as a wheelchair user's skills (e.g. wheelchair/transfer skills) [15], as well as their behavior/habits, social situation and activities should also be considered when developing fall prevention strategies with a wheelchair user [13]. As participants in this study suggested that fall prevention should be context and task-specific, home assessments may be an ideal time to practice transfer and wheelchair skills that are task and context-specific. Individualized wheelchair skills training conducted in the home environment improved the advanced wheelchair skills capacity of experienced community-dwelling veterans with SCI [39]. Home assessments need to be restructured to adequately address fall risk in this population. For instance, therapists should also review what activities a wheelchair user performs outside of their wheelchair and make appropriate safety recommendations. Moreover, it would be helpful to review the importance of regular equipment inspections (e.g. wheelchair's brakes, frame, control system) to reduce the risk of falls related to equipment failure [8, 46, 47].

Consistent with previous research, we found wheelchair users tended to fall while wheeling/propelling over uneven ground [3, 4, 7, 12]. The stability of a person's wheelchair, their satisfaction with their wheelchair, and their ergonomics while sitting in the wheelchair influenced the probability of a fall and fall-related injury. Yet, 55–68% of wheelchair users use inappropriate wheelchairs (e.g. inappropriate seat height, cushion and back height) due to factors such as users' obtaining wheelchairs without a prescription, wheelchairs were prescribed by non-specialized centres, errors in prescription and construction of wheelchairs, and challenges with insurances and refunding systems [48, 49]. In order to reduce/prevent fall risk and fall-related injuries, improvements in accessibility of wheelchair prescription and appropriate follow-up are needed for wheelchair users living in the community [48, 49].

The timing and duration of fall prevention should be tailored to each person's needs and preferences. When determining the best timing to deliver fall prevention, clinicians should consider a person's readiness to learn new information [50]. When it is offered too early, individuals may not be ready to learn the new information and the education may not be effective

[51]. Information related to community living may be more effective if it was introduced after re-integration to community living [51].

Effective education is a vital component of fall prevention [15]. Fall prevention should be developed based on adult learning principles that emphasize self-directed learning [52]. Participants in this study, as well as others, have requested the delivery of fall prevention information in a variety of different formats. This would accommodate different learning needs/preferences [15]. Also, fall prevention should also be ongoing as a person's condition or situation changes [13]. Community-dwellers with SCI preferred Internet-based health resources as these resources could be accessed at any time. Internet-based fall prevention resources could also be a solution for those that no longer have funding or access to consult a rehabilitation professional [15, 53]. However, the potential safety risks associated with learning skills from the Internet must be considered, as there is a lack of professional guidance and a possibility for misunderstanding the information [54].

Participants in the current study used technology-based devices, such as their smart home devices or voice-activated smart phones, to get assistance after a fall. Since smart phones and smart home devices are widely accessible to people in developed countries [55], clinicians should explore this as a potential strategy for management after a fall.

It is important to note that priorities for fall prevention that participants identified overlap with several strategies they already use [13], and this suggests that although these participants fell, they believe the strategies they personally use are effective and should be integrated into fall prevention. It may be possible that the new components (e.g. skills training and training from peers) that were suggested by these participants may potentially reduce their risk of falls. Another fall prevention priority was to educate and equip caregivers/family members/friends with fall prevention and management skills. Falls that occurred during assisted transfers were often a result of error by the caregiver/family member/friend; however, programs to educate formal and informal caregivers currently do not exist. It is possible that falls caused by formal caregiver errors may be related to a lack of training or time pressures.

We noted sex-differences in fall circumstances that should be explored in future studies. In our study females reported more falls at night than males. It is possible that this observation may be related to sex-based differences in urogenital physiology and/or toileting behaviours [56]. For instance, males are more likely to use bedside urinals than females [57]. Another interesting finding was that males did not report any falls while wheeling over uneven ground. According to previous literature, the weight of a wheelchair user can impact the wheelchair's stability and maneuverability [58]. One possible explanation for our finding is that males tend to weigh more than females [59] and due to heavier weight, males may be less likely to tip while going over uneven ground in a manual wheelchair. Sex/gender differences related to risk management strategies used may exist and this warrant further exploration.

The limitations of this study should be considered. First, the information collected in the surveys and focus group discussion was limited to the questions we had asked. Alternative questions could have yielded different information [3]. For instance, some survey responses were more relevant to ambulators with SCI. However, participants had an option to provide responses in the open text sections and the qualitative phase allowed us to expand discussions in these areas. Also, we did not ask about wheelchair and equipment-related behaviour such as whether participants were using their seat belts/chest straps at the time of the fall, or whether their wheelchair/equipment required any repairs [7]. We also neglected to ask about a person's environment set-up [3]. In retrospect, this information would have been valuable to understanding the circumstances of falls (e.g. whether living in a home with a small turning radius increases falls). Second, fall-related injuries in this study were self-reported. Self-reported data can under or over report information [60]. Third, since reminder phone calls were conducted

every three to four weeks, it is possible that fall recording might not be as accurate as data collection that involved more frequent phone calls. Fourth, a longer tracking period for falls (i.e. 12 months rather than 6 months) may have revealed a higher number of fallers [1]. Also, a Hawthorne effect may have occurred where prospective tracking of falls could have influenced the incidence proportions of falls [4, 61]. Fifth, our sample included a disproportionately higher number of manual wheelchair users as opposed to power wheelchair users. This impacted our findings as the circumstances of falls and fall prevention strategies primarily relate to manual wheelchair users. More research is needed to develop fall prevention strategies for those who use a power wheelchair. Lastly, the small sample size limits generalizability of the quantitative data [62].

Nevertheless, the prospective design and mixed methods are a methodological strength of this study [1]. Also, participants were followed by a single researcher, and this may have enhanced the participants' comfort during the focus group discussion. In addition, our use of a standardized definition of a fall ensured both the researchers and participants clearly understood what constituted as a fall for this study [63]. Integration of methods allowed us to gain an in-depth understanding of fall circumstances encountered by participants. There was also an unintended positive outcome of the photovoice focus group discussion where some participants were exposed to novel fall prevention strategies/tips introduced by other participants with SCI. Future research should investigate whether group-based and peer-led approaches to deliver fall prevention education would be feasible and effective in this population. Peer-based approaches to learning skills for people with spinal cord injury have previously been successful [64, 65], but have not been used for fall prevention [13].

In conclusion, participants foresee value in fall prevention initiatives that extend beyond what is currently offered. Findings from this study highlight suggestions and preferences for the structure and content of future fall prevention initiatives from the perspectives of wheelchair users with SCI. Moving forward, future research should explore the effectiveness of specific fall prevention initiatives for community-dwelling wheelchair users with SCI, while incorporating feedback from the target users to enhance retention in the programs.

## Supporting information

**S1 File. Fall survey.**
(DOCX)

**S2 File. Focus group interview guide.**
(DOCX)

**S3 File. Good reporting of a mixed methods study.**
(DOCX)

## Acknowledgments

We would like to acknowledge the participants for dedicating their time and sharing their experiences with us. We would like to thank the clinical staff at the Lyndhurst Centre for their assistance with the study recruitment.

## Author Contributions

**Conceptualization:** Carol Y. Scovil, Geoff Bostick, Anita Kaiser, B. Catharine Craven, Kristin E. Musselman.

**Data curation:** Hardeep Singh, Carol Y. Scovil, Anita Kaiser, Kristin E. Musselman.

**Formal analysis:** Hardeep Singh, Carol Y. Scovil, Geoff Bostick, Kristin E. Musselman.

**Funding acquisition:** Kristin E. Musselman.

**Investigation:** Hardeep Singh, Carol Y. Scovil, Kristin E. Musselman.

**Methodology:** Hardeep Singh, Carol Y. Scovil, Geoff Bostick, B. Catharine Craven, Susan B. Jaglal, Kristin E. Musselman.

**Project administration:** Hardeep Singh, Kristin E. Musselman.

**Resources:** Hardeep Singh, Anita Kaiser, Kristin E. Musselman.

**Software:** Kristin E. Musselman.

**Supervision:** Susan B. Jaglal, Kristin E. Musselman.

**Validation:** Carol Y. Scovil, Geoff Bostick, Anita Kaiser, Kristin E. Musselman.

**Visualization:** Hardeep Singh, Kristin E. Musselman.

**Writing – original draft:** Hardeep Singh, Kristin E. Musselman.

**Writing – review & editing:** Hardeep Singh, Carol Y. Scovil, Geoff Bostick, Anita Kaiser, B. Catharine Craven, Susan B. Jaglal, Kristin E. Musselman.

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
