## [Decision Letter · Decision Letter 0]

20 Jul 2020

PONE-D-20-03903

Perspectives of wheelchair users with spinal cord injury on fall circumstances and fall prevention: a mixed methods approach using photovoice

PLOS ONE

Dear Dr. Musselman,

Thank you for submitting your manuscript to PLOS ONE. After careful consideration, we feel that it has merit but does not fully meet PLOS ONE’s publication criteria as it currently stands. Therefore, we invite you to submit a revised version of the manuscript that addresses the points raised during the review process.

We look forward to receiving your revised manuscript.

Kind regards,

Yih-Kuen Jan, PhD

Academic Editor

PLOS ONE

Journal Requirements:

2. Thank you for including your funding statement; "The funders had no role in study design, data collection and analysis, decision to publish, or preparation of the manuscript."

Reviewers' comments:

Reviewer's Responses to Questions

**Comments to the Author**

1. Is the manuscript technically sound, and do the data support the conclusions?

Reviewer #1: Yes

Reviewer #2: Yes

2. Has the statistical analysis been performed appropriately and rigorously? 

Reviewer #1: Yes

Reviewer #2: Yes

3. Have the authors made all data underlying the findings in their manuscript fully available?

Reviewer #1: Yes

Reviewer #2: Yes

4. Is the manuscript presented in an intelligible fashion and written in standard English?

Reviewer #1: Yes

Reviewer #2: Yes

5. Review Comments to the Author

Reviewer #1: I thank the editor for allowing me to rate this article.

The value is well conducted and well detailed in all its points. It has a strong and precise methodological structure.

the evaluation of aids in patients with spinal cord injury is very important, but this work certainly makes a good contribution to the scientific literature

I therefore request that the article be accepted in its current form without modification.

Reviewer #2: Thank you for the opportunity to review your manuscript entitled “Perspectives of wheelchair users with spinal cord injury on fall circumstances and fall prevention: a mixed methods approach using photovoice”. The manuscript is clear, very well written, the topic would be of interest of readers, the manuscript would also contribute to develop ‘effective’ fall prevention programs among wheelchair users with SCI and improve the quality of care in this population. I have highlighted some points that I believe would improve the quality of the manuscript.

Reference 13 is not under review anymore. Please update this!

Figure 1 and line 153: 1-year fall tracking has shown a higher frequency of fall among wheelchair users with SCI. It is not clear why the authors decided a 6-month fall tracking? Please provide a justification.

Line 137: Were wheelchair athletes included? For example: a fall during a wheelchair basketball game is different from a fall which results from a wheelchair to bed fall. This needs to be addressed. During a wheelchair basketball game, a wheelchair user can fall 5 or more times. Did you ask participant to count those falls in this study?

Line 160: 3 to 4 weeks before reminding participants to complete the fall survey seems a lot and the fall recording might be not that accurate (generally 2 weeks is used). This should be addressed in the limitations of the study.

Line 229: The difference is just significant, a priori; this is almost p = 0.05

Table 2: A better description is needed for falls from sports/exercise.

I suggest also exploring the relationship between recurrent falls (more than 1 and report of injuries). This might consolidate the existing literature.

It is surprising that ‘Environment’ was not found as a subtheme since slippery surface, cluttered space, etc. have been frequently reported as fall risk factors as well among wheelchair users with SCI. This does not necessarily need to be in the community but at home.

I was expecting to see any specificity for manual wheelchair users vs power wheelchair users in terms of fall circumstances and fall prevention recommendations (even though your sample presented with few power wheelchair users). This could provide some information for future research.

Lines 597 – 601: Another explanation might be the difference in risk management; women tend to take less risk than men and might avoid going through uneven surfaces. This could definitely limit them to do some activities.

6. PLOS authors have the option to publish the peer review history of their article (what does this mean?). If published, this will include your full peer review and any attached files.

Reviewer #1: No

Reviewer #2: No

---

## [Author Response · Author response to Decision Letter 0]

29 Jul 2020

We have addressed each comment from the Editor and Reviewers below. Each comment from the Editor/Reviewers is pasted first, and our response follows. The requested changes are tracked in the manuscript document.

Comments from the Editor

Author’s response: Thank you for providing the style requirements. As per the requirements, we have renamed the files and changed the heading styles.

2. Thank you for including your funding statement; "The funders had no role in study design, data collection and analysis, decision to publish, or preparation of the manuscript." At this time, please address the following queries: Please clarify the sources of funding (financial or material support) for your study. List the grants or organizations that supported your study, including funding received from your institution. State what role the funders took in the study. If the funders had no role in your study, please state: “The funders had no role in study design, data collection and analysis, decision to publish, or preparation of the manuscript.” If any authors received a salary from any of your funders, please state which authors and which funders.If you did not receive any funding for this study, please state: “The authors received no specific funding for this work.” Please include your amended statements within your cover letter; we will change the online submission form on your behalf.

Author’s response: The following sentence has been added to the cover letter: “This study was funded by a Craig H. Neilsen Foundation Psychosocial Research Grant to KEM, a Canadian Institutes of Health Research Fellowship to HS and Toronto Rehabilitation Institute Student Scholarship to HS. SBJ holds the Toronto Rehabilitation Institute Chair-University Health Network at the University of Toronto. The funders did not have a role in the study’s design, data collection and analysis, decision to publish or preparation of the manuscript.” 

In your revised cover letter, please address the following prompts: a) If there are ethical or legal restrictions on sharing a de-identified data set, please explain them in detail (e.g., data contain potentially identifying or sensitive patient information) and who has imposed them (e.g., an ethics committee). Please also provide contact information for a data access committee, ethics committee, or other institutional body to which data requests may be sent.

We have added the following sentence to the cover letter: “Data Availability statement: The datasets generated and/or analysed during the current study are not publicly available due to the qualitative data (i.e. transcripts) containing information that could compromise research participant privacy/consent and restrictions by the research ethics committee. Data requests can be made to the University Health Network’s Research Ethics Board at reb@uhnresearch.ca or 416-581-7849 (reference 14-8569-DE).” 

Author’s response: We have added an ORCID iD for the corresponding author Dr. Kristin Musselman (ORCID iD: https://orcid.org/0000-0001-8336-8211).

Reviewer 2’s Comments to Author:

1. Reference 13 is not under review anymore. Please update this!

Author’s response: Thank you for highlighting this. We have updated Reference 13.

2. Figure 1 and line 153: 1-year fall tracking has shown a higher frequency of fall among wheelchair users with SCI. It is not clear why the authors decided a 6-month fall tracking? Please provide a justification.

Author’s response: In 2019, a meta-analysis published by our research team reported that “researchers should be aware that the duration of monitoring will influence the incidence proportion of falls. As longitudinal studies lasting 12 months are not always feasible, we recommend a minimum monitoring period of 6 months, which would enable comparison of one’s data with previously documented rates” (Khan et al. 2019). We chose to track falls for 6-months in the current study as this duration of fall monitoring was feasible to complete in the two years of grant funding received for this project. Further, this duration met the minimum duration recommended (Khan et al. 2019) and enables comparison of the data presented here to previous research. We have added this as a limitation of our study (lines 624-625): “a longer tracking period for falls (i.e. 12 months rather than 6 months) may have revealed a higher number of fallers [1]”. 

3. Line 137: Were wheelchair athletes included? For example: a fall during a wheelchair basketball game is different from a fall which results from a wheelchair to bed fall. This needs to be addressed. During a wheelchair basketball game, a wheelchair user can fall 5 or more times. Did you ask participant to count those falls in this study?

Author’s response: Merriam-webster defines an athlete as “a person who is trained or skilled in exercises, sports, or games requiring physical strength, agility, or stamina.” Based on this broad definition of an athlete, it is possible that there were individuals included that were skilled in sports; however, they engaged in sports for recreation and were not full-time “athletes”. While some studies have excluded falls during sports from their analysis (Forslund et al., 2017), we believe it is important to include these falls because these would count as falls as per the definition of falls we used in this study: “an event which results in a person coming to rest inadvertently on the ground or floor or other lower level.” Based on this definition, falls during sports/exercise should be included. Furthermore, sports are part of a meaningful recreational activity and we were interested in examining falls during all activities a wheelchair user engaged in rather than select activities. 

However, we can appreciate that these falls are quite different than a fall during a transfer. As such, we have separated these types of falls in different categories in table 2 (“activity during fall”). We have provided more details about the falls that occurred during sports/exercise in the results (lines 244-246): “Sports/exercise included activities such as “exercising from my chair”, “catching a frisbee” and “playing wheelchair basketball”.”

4. Line 160: 3 to 4 weeks before reminding participants to complete the fall survey seems a lot and the fall recording might be not that accurate (generally 2 weeks is used). This should be addressed in the limitations of the study.

Author’s response: We have acknowledged this as a limitation of our study (lines 622-624): “since reminder phone calls were conducted every three to four weeks, it is possible that fall recording might not be as accurate as data collection that involved more frequent phone calls.”

5. Line 229: The difference is just significant, a priori; this is almost p = 0.05

Author's Response: Yes, we agree with the reviewer’s interpretation of this result, and feel that our phrasing of the result is consistent with this interpretation (lines 228-229): “Fallers tended to be younger…”

6. Table 2: A better description is needed for falls from sports/exercise.

Author’s response: As recommended, we have provided more details about falls from sports/exercise: (lines 244-246): “Sports/exercise included activities such as “exercising from my chair”, “catching a frisbee” and “playing wheelchair basketball”.”

7. I suggest also exploring the relationship between recurrent falls (more than 1 and report of injuries). This might consolidate the existing literature.

Author’s response: We have added the following information regarding recurrent falls and report of injuries (lines 269-272): “Ten fall-related injuries were reported by nine participants who experienced one or more falls that resulted in an injury, with the most common injuries being cuts/scrapes and pain. . Of these nine participants, five were recurrent fallers and four fell only one time over the six-month period.”

8. It is surprising that ‘Environment’ was not found as a subtheme since slippery surface, cluttered space, etc. have been frequently reported as fall risk factors as well among wheelchair users with SCI. This does not necessarily need to be in the community but at home.

Author’s response: We agree this was interesting. In the qualitative discussions, participants did not associate their home environment as “risky”; they considered their home “a safe space” (subtheme 2 under theme 1). This was also why participants tended to be “carelessness, multi-tasking, and not paying attention to the surroundings” as they did not consider the home environment to be dangerous. As one participant explained, “usually when we are in the home, we tend to be careless…I pay less attention to my surroundings when I am at home because it is a ‘safe space.” 

9. I was expecting to see any specificity for manual wheelchair users vs power wheelchair users in terms of fall circumstances and fall prevention recommendations (even though your sample presented with few power wheelchair users). This could provide some information for future research.

Author’s response: Fall circumstances for manual wheelchair users vs power wheelchair users are reported separately in table 2. In terms of recommendations for future research. As we only have few power wheelchair users, we do not have enough data on power wheelchair users to make any recommendations, other than more research is needed for those who use a power wheelchair (lines 626-630): “our sample included a disproportionately higher number of manual wheelchair users as opposed to power wheelchair users. This impacted our findings as the circumstances of falls and fall prevention strategies primarily relate to manual wheelchair users. More research is needed to develop fall prevention strategies for those who use a power wheelchair.”

10. Lines 597 – 601: Another explanation might be the difference in risk management; women tend to take less risk than men and might avoid going through uneven surfaces. This could definitely limit them to do some activities.

Author’s response: We found falls while going over uneven surfaces were reported by females and an interesting finding was that males did not report any falls while wheeling over uneven ground. We agree there may be differences in risk management approaches used by men and women. This has been added as a future direction to lines 609-610: “Sex/gender differences related to risk management strategies used may exist and this warrant further exploration.”

---

## [Decision Letter · Decision Letter 1]

11 Aug 2020

Perspectives of wheelchair users with spinal cord injury on fall circumstances and fall prevention: a mixed methods approach using photovoice

PONE-D-20-03903R1

Dear Dr. Musselman,

We’re pleased to inform you that your manuscript has been judged scientifically suitable for publication and will be formally accepted for publication once it meets all outstanding technical requirements.

Kind regards,

Yih-Kuen Jan, PhD

Academic Editor

PLOS ONE

Additional Editor Comments (optional):

Reviewers' comments:

Reviewer's Responses to Questions

**Comments to the Author**

1. If the authors have adequately addressed your comments raised in a previous round of review and you feel that this manuscript is now acceptable for publication, you may indicate that here to bypass the “Comments to the Author” section, enter your conflict of interest statement in the “Confidential to Editor” section, and submit your "Accept" recommendation.

Reviewer #2: All comments have been addressed

2. Is the manuscript technically sound, and do the data support the conclusions?

Reviewer #2: Yes

3. Has the statistical analysis been performed appropriately and rigorously? 

Reviewer #2: Yes

4. Have the authors made all data underlying the findings in their manuscript fully available?

Reviewer #2: (No Response)

5. Is the manuscript presented in an intelligible fashion and written in standard English?

Reviewer #2: Yes

6. Review Comments to the Author

Reviewer #2: (No Response)

7. PLOS authors have the option to publish the peer review history of their article (what does this mean?). If published, this will include your full peer review and any attached files.

Reviewer #2: No

---

## [Editor Report · Acceptance letter]

17 Aug 2020

PONE-D-20-03903R1 

Perspectives of wheelchair users with spinal cord injury on fall circumstances and fall prevention: a mixed methods approach using photovoice 

Dear Dr. Musselman:

I'm pleased to inform you that your manuscript has been deemed suitable for publication in PLOS ONE. Congratulations! Your manuscript is now with our production department. 

Kind regards, 

on behalf of

Dr. Yih-Kuen Jan 

Academic Editor

PLOS ONE